# Further characterization of the effect of the prototypical antidepressant imipramine on the microstructure of licking for sucrose

**Paolo S. D'Aquila**⊙*, **Adriana Galistu**

Dipartimento di Scienze Biomediche, Università di Sassari, Sassari, Italy

* dsfpaolo@uniss.it

## Abstract

We previously reported that treatment with the prototypical antidepressant imipramine induced a dose-dependent reduction of the ingestion of a 10% sucrose solution, due to reduction of the licking burst number, thus suggesting reduced motivation and/or increased satiation. Importantly, the experimental sessions were performed in an alternate order, either 1-h or 24-h after imipramine administration. The observation that imipramine effect was more pronounced in the "1-h after-treatment" sessions, i.e. at the time of the brain drug $C_{max}$, led us to suggest that it was likely related to brain drug levels at testing time. However, such an experimental design does not allow to rule out the alternative possibility that the observed effect might be due to post-session administration, as previously observed with memantine. To determine whether imipramine-induced decrease of sucrose ingestion could be observed even in absence of post-session administration, we examined the effect of a daily 22 day treatment with imipramine (5, 10 and 20 mg/kg). In the first half of the treatment period all behavioural tests were performed 1-h after administration. In the second half of the treatment period, tests were performed alternatively either 1-h or 24-h after imipramine administration. The results confirm that imipramine reduces sucrose ingestion due to a reduction of the licking burst number. Most importantly, these results demonstrate that this effect does not require imipramine post-session administration, since it was present before the beginning of post-session administrations. This supports the interpretation of the reduction of sucrose ingestion as a consequence of reduced motivation and/or increased satiation. Thus, these findings, taken together with the results of our previous study, might be relevant in explaining the effects of imipramine in models of drug-seeking and in body weight gain reduction in rats, but not in accounting for the antidepressant therapeutic effect. At variance with the results of our previous study, an increase in burst size was present in the first half of the treatment period, which might be interpreted as a prohedonic effect and/or as a compensatory effect.

## Introduction

Experimental measures provided by the analysis of licking microstructure might be interpreted within the framework of psychological constructs relevant to depression, such as

**Data Availability Statement:** All relevant data are within the manuscript and its Supporting Information files.

**Funding:** This work was supported by the Fondazione di Sardegna (U500.2013/AI.424.MGB

to PSD) and by the University of Sassari Research Fund 2020 (PSD). The funders had no role in study design, data collection and analysis, decision to publish, or preparation of the manuscript.

**Competing interests:** The authors have declared that no competing interests exist.

pleasure–hedonic impact/reward evaluation–, and motivation–incentive salience attribution/ behavioural activation [1–12]. Rats ingesting fluids emit bursts of licks with a frequency of five to seven licks *per* second. The size of licking bursts–heretofore burst size–corresponds to the number of licks *per* burst and is particularly sensitive to stimuli involving the orosensory contact with the reward, such as taste stimuli. This measure was suggested to represent pleasure. The number of licking bursts–heretofore burst number–corresponds to the number of times that the subjects initiate licking behaviour and is particularly sensitive to stimuli, such as post-ingestional cues, which do not involve the orosensory contact with the reward. (But notice that this measure does respond to orosensory cues, albeit in a less straightforward fashion than burst size.) Burst number was suggested to represent motivation [4–12].

Monitoring sucrose consumption and preference in rats (and mice) is a widely used method in the assessment of the sensitivity to reward in behavioural paradigms aimed to model aspects of depression. In models such as chronic mild stress (CMS), repeated restraint stress and maternal deprivation, exposure of the experimental subjects to a variety of stressors yields a reduction of the preference for and of the consumption of sweet solutions. This effect is interpreted as "anhedonia"–inability to experience pleasure, a core symptom of depression– and is reversed by antidepressant drugs [2,3].

The acute effect of treatment with a few antidepressant drugs was investigated in a small number of studies [13–16]. Recently, we reported that treatment with the prototypical antide-pressant imipramine [17] yielded a dose-dependent reduction of the ingestion of a 10% sucrose solution. Since this effect was accounted for by a reduction of burst number [18], it was interpreted as reduced motivation and/or increased satiation. In more detail, ten experi-mental sessions were performed in an alternate order, either 1-h or 24-h after imipramine administration, in the course of a daily 21 day treatment. The observation that imipramine effect was more pronounced in the "1-h after-treatment" sessions, i.e. at the time of the brain drug $C_{max}$ [19], led us to suggest that it was likely related to brain drug levels at testing time [18]. However, the design of this study does not allow to rule out the alternative possibility that the observed effect might be due to post-session administration. Indeed, we have recently observed that repeated post-session administration of memantine induced a decrease of the activation of licking behaviour, indicated by reduced burst number [20]. Notably, if the experi-mental sessions were performed in an alternate order either 1-h or 24-h after memantine administration [21]–as in the cited study with imipramine [18]–the effect was similar to the effect observed with memantine post-session administration [20], and–as observed with imip-ramine–was more pronounced in the 1-h "after-treatment" sessions [21]. As for the interpreta-tion of this observation, we suggested the possibility of the development of a memory-related effect, such as conditioned taste-aversion [20,21]. However, this question remains to be settled by further experiments.

The aim of this study was to determine whether imipramine-induced decrease of sucrose ingestion could be observed even in absence of post-session administration. To this end, we examined the effect of imipramine in the course of a daily treatment for 22 days. In the first half of the treatment period all behavioural tests were performed 1-h after administration. In the second half of the treatment period, tests were performed alternatively either 1-h or 24-h after imipramine administration (Table 1).

## Materials and methods

### Subjects and drug treatments

Experimentally naïve male Sprague-Dawley rats (Harlan, Italy) aged about 10 weeks and weighing 300–350 g at the beginning of the experiment were used as subjects. The animals

**Table 1. Experiment timeline.**

| Treatment and session timeline | | | | | | | | | | | | | | | | | | | | | | |
|---|---|---|---|---|---|---|---|---|---|---|---|---|---|---|---|---|---|---|---|---|---|---|
| Part 1 | | | | | | | | | | | Part 2 | | | | | | | | | | | |
| 1 | 2 | 3 | 4 | 5 | 6 | 7 | 8 | 9 | 10 | 11 | 12 | 13 | 14 | 15 | 16 | 17 | 18 | 19 | 20 | 21 | 22 | 23 |
| ↓* | ↓ | ↓ | ↓ | ↓* | ↓ | ↓ | ↓* | ↓ | ↓ | ↓ | *↓ | ↓ | ↓ | ↓* | ↓ | *↓ | ↓ | ↓* | ↓ | ↓ | ↓* | * |

Numbers in the first row indicate consecutive days. Asterisks indicate experimental sessions, arrows indicate imipramine injections, performed either before or after–left and right with respect to the asterisk, respectively–experimental sessions.

were housed in groups of two-three per cage in controlled environmental conditions (temperature 22–24° C; humidity 50–60%; light on at 08:00, off at 20:00), with free access to food and water. Imipramine HCl (Sigma, St Louis, USA) was dissolved in distilled water and injected intra-peritoneally (i.p.) in a volume of 1 ml/kg. Vehicle treatment consisted in a 1 ml/kg distilled water i.p. administration.

## Ethics statement

All the experimental procedures were carried out in accordance with the regulatory requirement of the Italian law (D.L. 116, 1992) and Council Directive 2010/63EU of the European Parliament and Council, and were approved by the Istituto Superiore di Sanità (protocol n. 4211/2014) and authorised by the Ministry of Health, Italy (protocol n. 74/2014-B). At the end of the experiment, the rats were euthanised with pentobarbital sodium. Animals were monitored and properly handled throughout the experiment, and every effort was made to minimize suffering or pain.

## Apparatus, microstructural measures and testing conditions

Behavioural testing was carried out using a multistation lick analysis system (Habitest, Coulbourn Instruments, USA) connected to a computer. Rats were individually placed in a Perspex chamber with an opening in the centre of the front wall allowing access to a bottle spout. The recording period started either after the first lick or after 3-min that the animals were placed into the chambers, so that the latency to the first lick had a cut off time of 3-min. The interruptions of a photocell beam by each single tongue movement while licking the spout were recorded, with a temporal resolution to the nearest 20 milliseconds. The raw data were analysed through Graphic State 3.2 software (Coulbourn Instruments, USA) and, besides lick number, the following microstructural measures were obtained: number of bursts, time spent in bursts, latency to the first lick. A burst was defined as a series of licks with pauses no longer than 400 milliseconds. This pause criterion was adopted in our lab [18,22] because it is just longer than the break point of a log-survivor plot of inter-lick intervals [23]. Burst size (number of licks *per* burst) and intra-burst lick rate (lick/sec within bursts) were then calculated. The data were collected in time bins of 3-min in sessions of 30-min.

The experiments were performed between 09:00 and 13:00, i.e. during the light phase of the lighting cycle. All the experiments were perfomed in non-deprived animals.

## Procedures

The subjects were first familiarised with the test apparatus in 30-min training sessions where they had access to a 10% sucrose solution. Based on the whole-session mean burst size of the last training session, they were allocated into four matched groups to be treated with daily i.p. injections of either vehicle or one of three doses of imipramine: 5, 10 and 20 mg/kg. The

training sessions were 9 in 15 days. Food and water were continuously available to the subjects between training/experimental sessions. The 10% sucrose solution was made available only during training/experimental sessions. Starting three days after the last training session, imipramine (or its vehicle) was administered daily for 22 days to 35 subjects (vehicle, n = 9; imipramine: 5 mg/kg, n = 8; 10 mg/kg, n = 9; 20 mg/kg, n = 9). Nine 30-min experimental sessions were performed either 1-h after treatment–in the treatment days 1, 5, 8, 15, 19 and 22 –or 24-h after treatment–in the treatment days 12, 17 and in the first day after the last treatment (day 23). This implies that in days 12 and 17 the animals were treated immediately after the testing session (Table 1). One subject treated with the dose of imipramine 20 mg/kg was withdrawn from the experiment due to health problems before the end of imipramine treatment.

## Statistical analysis

Statistical analysis of all sets of data was performed with ANOVA, by the software Statistica 8.0 (StatSoft Inc.). When a significant main effect of the factor *dose* was present, post-hoc comparisons between each imipramine dose-treated group and the vehicle-treated group were made by the Dunnett's test. When a significant interaction between factors was revealed, comparisons were performed by F-test for contrasts [24,25].

Body weight data were analysed by ANOVA, with *dose* as a between-group factor, with four levels corresponding to vehicle and the three imipramine doses, and *time* as a within-group factor, with two levels corresponding to the start and the end of treatment.

ANOVA of the whole-session data of the first part of the experiment (days 1, 5 and 8) involved *dose* as a between-group factor and *session* as a within-group factor. The analysis of the whole-session data of the second part of the experiment–i.e. days 15, 19, 22 with sessions performed 1-h after treatment and days 12, 17, 23 with sessions performed 24-h after treatment–involved *administration time* as a further within-group factor. An additional analysis of the number of licks *per* burst data limited to the first 3-min time bin was performed.

The data of the within-session burst number time-course relative to the sessions performed in the first part of the experiment (days 1, 5, 8) were analysed by ANOVA, with *dose* as a between-group factor and *time* as a within-group factor, with 10 levels corresponding to the ten 3-min time bins within the session.

# Results

## Body weights

F-tests for contrasts performed on the basis of a statistically significant *dose×time* interaction [$F_{(3,30)} = 45.94$, $P < 10^{-6}$] showed that, at the end of treatment, the average body weight of the groups treated with vehicle and imipramine 5 mg/kg were increased with respect to the start treatment values, the dose of 10 mg/kg prevented body weight gain, and the dose of 20 mg/kg resulted in a significant weight loss. As a result, the average body weight of the groups treated with 10 and 20 mg/kg were reduced with respect to the vehicle-treated group (Table 2).

## Time-course across sessions of the effect of imipramine on licking microstructure

ANOVA of the whole-session lick number data of the first three sessions performed 1-h after treatment–i.e. the first part of the experiment–failed to show a statistically significant *dose×session* interaction [$F_{(6,62)} = 2.09$, n.s.]. However, a significant main effect of *dose* was present [$F_{(3,31)} = 3.4$, $P = 0.029$]. Dunnett's test revealed a significant decrease of lick number with respect to the vehicle-treated group in the group treated with imipramine 20 mg/kg, regardless

**Table 2. Body weights.**

| IMI mg/kg | Start treatment | End treatment |
|---|---|---|
| 0 | 353.88±5.93 | 393.88±8.19[++] |
| 5 | 351.25±7.89 | 386.87±9.99[++] |
| 10 | 356.66±7.07 | 351.11±10.06[*] |
| 20 | 362.22±10.74 | 330.62±10.32[***+] |

IMI: Imipramine. With respect to the corresponding vehicle treatment value

[*]P<0.01

[**]P<0.0001; with respect to the start treatment value of the corresponding group

[+]P<10^{-4}

[++]P<10^{-6} (ANOVA followed by F-tests for contrasts).

of session (Fig 1, top left panel). ANOVA of the whole-session lick number data of the second part of the experiment–days 15, 19, 22 performed 1-h after treatment and days 12, 17, 23 performed 24-h after treatment–revealed a significant effect of the factor *dose* [F(3,30) = 3.89; P = 0.018] which was due to a reduction of lick number with the dose of 20 mg/kg, regardless of session and administration time [*dose×administration time*: F(3,30) = 0.42, n.s.; *dose×session*: F(6,60) = 1.36, n.s.; *dose×administration time×session*: F(6,60) = 2.2, n.s.] (Fig 1, top panels).

ANOVA of the whole-session burst number data of the first three sessions performed 1-h after treatment–i.e. the first part of the experiment–failed to show a statistically significant *dose×session* interaction [F(6,62) = 1.002, n.s.]. However, a significant main effect of *dose* was present [F(3,31) = 6.12, P = 0.0021]. Dunnett's test revealed a significant decrease of burst number with respect to the vehicle-treated group in the groups treated with the three imipramine doses, regardless of session (Fig 1, mid left panel). ANOVA of the whole-session burst number data of the second part of the experiment–days 15, 19, 22 performed 1-h after treatment and days 12, 17, 23 performed 24-h after treatment–revealed a significant *dose×administration time* interaction [F(3,30) = 3.89; P = 0.018]. F-tests based on this interaction–i.e. comparisons between doses regardless of session in the two administration time conditions, 1-h and 24-h after treatment–showed a significant reduction of burst number with respect to the vehicle-treated group in the group treated with 10 mg/kg, both in the sessions performed 1-h after treatment and in the sessions performed 24-h after treatment (Fig 1, mid panels).

ANOVA of the whole-session number of licks *per* burst data (burst size) of the first three sessions performed 1-h after treatment–i.e. the first part of the experiment–failed to show a statistically significant *dose×session* interaction [F(6,58) = 1.88, n.s.]. However, a significant main effect of *dose* was present [F(3,29) = 3.56, P = 0.026]. Dunnett's test revealed a significant increase of burst size with respect to the vehicle-treated group in the group treated with the dose of imipramine 10 mg/kg, regardless of session (Fig 1, bottom left panel). ANOVA of the whole-session burst size data of the second part of the experiment–days 15, 19, 22 performed 1-h after treatment and days 12, 17, 23 performed 24-h after treatment–revealed a significant *dose×administration time* interaction [F(3,27) = 4.59; P = 0.01]. However, F-tests based on this interaction–i.e. comparisons between doses regardless of session in the two administration time conditions, 1-h and 24-h after treatment–failed to show any significant differences between the groups (Fig 1, bottom panels).

Analysis of the early session burst size (limited to the first 3-min time bin) yielded very similar results (S1 Fig). ANOVA of the first three sessions performed 1-h after treatment–i.e. the first part of the experiment–failed to show a statistically significant *dose×session* interaction [F

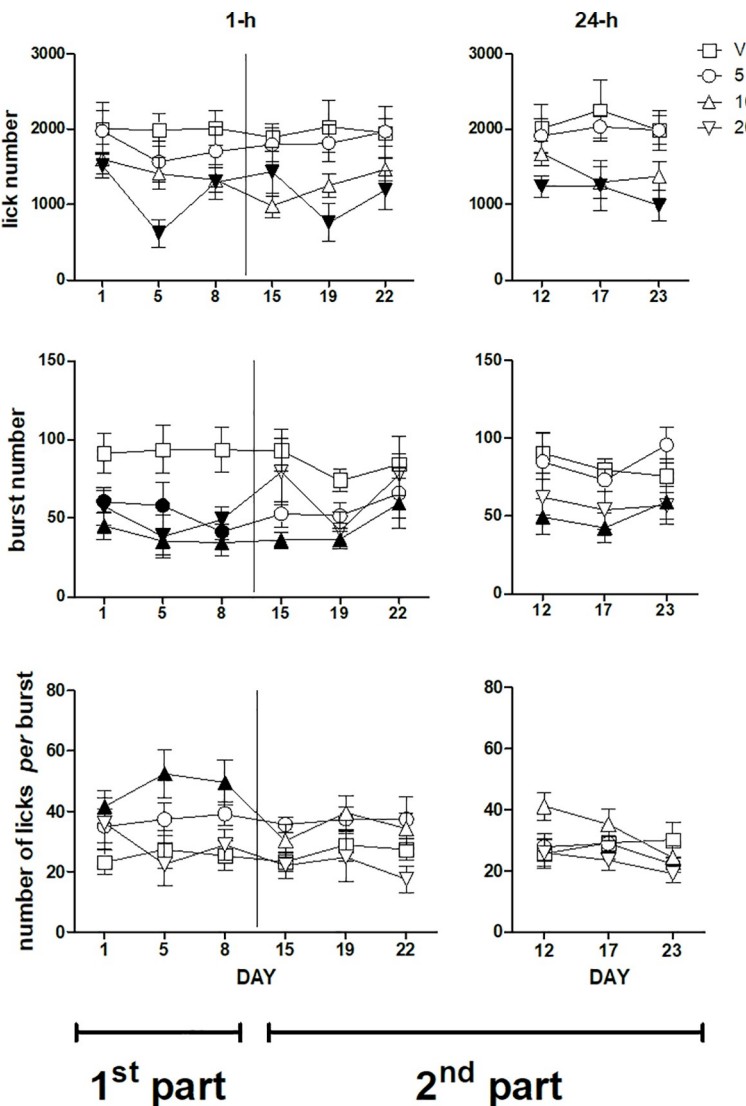

**Fig 1.** Lick number (top panels), burst number (mid panels) and number of licks *per* burst (bottom panels) during a 22 day daily treatment with imipramine. Left and right panels report, respectively, the data from the sessions performed 1-h and 24-h after imipramine administration. A vertical line in the left panels separates the first from the second part of the experiment. The data of the first part (days 1, 5, 8, left panels) and of the second part of the experiment (days 15, 19, 22, left panels, along with days12, 17, 23, right panels) were subjected to two separate analyses. Values represent the mean ± S.E.M. from 8–9 subjects. Black symbols indicate a statistically significant difference (P<0.05) with respect to the corresponding value of the vehicle-treated group (ANOVA followed by F-test for contrasts or Dunnett's test).

(6,58) = 1.16, n.s.]. However, a significant main effect of *dose* was present [F(3,03) = 3.56, P = 0.044]. Dunnett's test revealed a significant increase of burst size with respect to the vehicle-treated group in the group treated with the dose of imipramine 10 mg/kg, regardless of session. ANOVA of the data of the second part of the experiment–days 15, 19, 22 performed 1-h after treatment and days 12, 17, 23 performed 24-h after treatment–failed to show any statistically significant effect (S1 Fig).

S2 Fig reports the within-session time-course of burst size of the first three sessions (first part of the experiment). The data were not subjected to statistical analysis due to the numerous empty cells, indicating absence of licking behaviour within a time bin. Consistently with

previous evidence (e.g. [4,22,26]), the number of subjects failing to lick is particularly high late in the session–and in the groups treated with the two highest doses of imipramine (see S1 Data). Therefore, differences between time bin values early and late in the session are heavily affected by differences between subjects. Bearing in mind these limitations, the inspection of the graphs does not suggest relevant changes of burst size in the course of the session.

ANOVA of the whole-session intra-burst lick rate data (licks/sec within bursts) of the first three sessions performed 1-h after treatment–i.e. the first part of the experiment–failed to show a statistically significant *dose×session* interaction [F(6,58) = 1.024, n.s.]. However, a significant main effect of *dose* was present [F(3,29) = 7.17, P = 0.00095]. Dunnett's test revealed a significant decrease of intra-burst lick rate with respect to the vehicle-treated group in the group treated with the dose of imipramine 20 mg/kg, regardless of session (Fig 2, top left panel). ANOVA of the whole-session intra-burst lick rate data of the second part of the experiment–days 15, 19, 22 performed 1-h after treatment and days 12, 17, 23 performed 24-h after treatment–revealed a significant *dose×administration time* interaction [F(3,27) = 3.45; P = 0.03]. F-tests based on this interaction–i.e. comparisons between doses regardless of session in the two administration time conditions, 1-h and 24-h after treatment–showed a reduced intra-burst lick rate in the groups treated with 10 and 20 mg/kg imipramine in the sessions performed 1-h after drug administration, while in the sessions performed 24-h after drug administration the decrease in this measure was observed only in the group treated with 20 mg/kg (Fig 2, top panels).

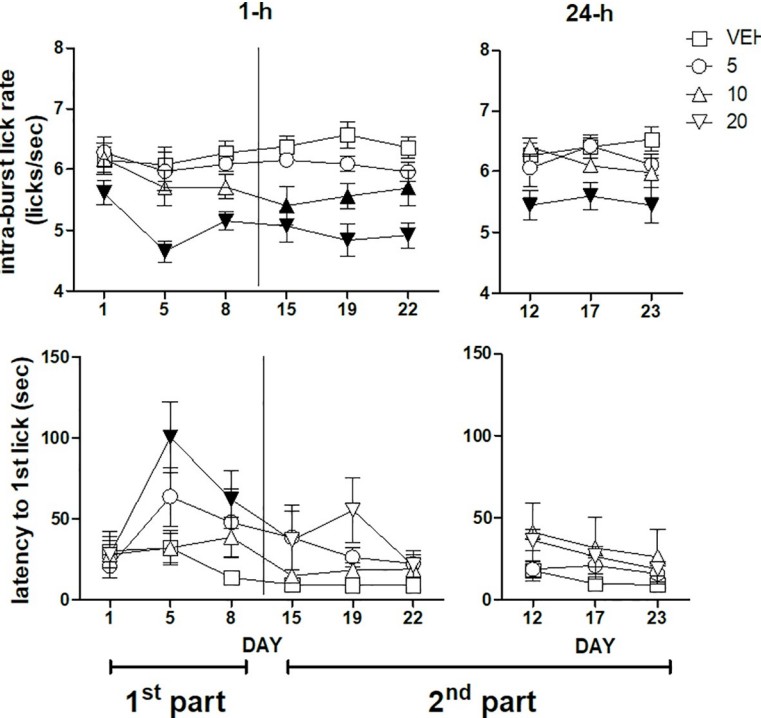

**Fig 2.** Intra-burst lick rate (top panels) and latency to the first lick (bottom panels) during a 22 day daily treatment with imipramine. Left and right panels report, respectively, the data from the sessions performed 1-h and 24-h after imipramine administration. A vertical line in the left panels separates the first from the second part of the experiment. The data of the first part (days 1, 5, 8, left panels) and of the second part of the experiment (days 15, 19, 22, left panels, along with days12, 17, 23, right panels) were subjected to two separate analyses. Values represent the mean ± S.E.M. from 8–9 subjects. Black symbols indicate a statistically significant difference (P<0.05) with respect to the corresponding value of the vehicle-treated group (ANOVA followed by F-test for contrasts or Dunnett's test).

ANOVA of the latency to the first lick data of the first three sessions performed 1-h after treatment–i.e. the first part of the experiment–showed a statistically significant *dose×session* interaction [$F(6,62) = 2.33$, $P = 0.042$]. F-test for contrasts showed an increased latency in the second and in the third session (day 5 and 8) in the group treated with imipramine 20 mg/kg (Fig 2, bottom left panel). ANOVA of the latency data of the second part of the experiment–days 15, 19, 22 performed 1-h after treatment and days 12, 17, 23 performed 24-h after treatment–failed to show any statistically significant effect (Fig 2, bottom panels).

## Within-session time-course of imipramine effect on burst number in the three sessions performed in the first part of the experiment

ANOVA of the data of the first session (day 1) showed a statistically significant effect of the factor *dose* [$F(3,31) = 3.45$, $P = 0.02$], due to the reduced burst number for the whole session observed with the dose of 10 mg/kg (Dunnett's test). A statistically significant effect of the factor *time* was also present [$F(9,279) = 24.01$, $P<10^{-6}$], due to the within-session decline observed regardless of *dose*. The interaction between the two factors was not statistically significant [$F(27,279) = 0.97$, n.s.] (Fig 3, top panel).

ANOVA of the data of the second session (day 5) showed a statistically significant *dose×time* interaction [$F(27,279) = 1.58$, $P = 0.03$]. Treatment with the dose of 5 mg/kg imipramine resulted in a reduced level of burst number in the 12–15-min time bin. The dose of 10 mg/kg reduced burst number level for the first 18-min of the session and in the last 3 time bins (21–30-min). Finally, treatment with the dose of 20 mg/kg reduced burst number in the first 21-min of the session and in the last time bin (27–30-min) (Fig 3, mid panel).

ANOVA of the data of the third session (day 8) showed a statistically significant effect of the factor *dose* [$F(3,31) = 7.58$, $P = 0.0006$], due to the reduced burst number for the whole session observed with all the three doses of imipramine (Dunnett's test). A statistically significant effect of the factor *time* was also present [$F(9,279) = 17.06$, $P<10^{-6}$], due to the within-session decline observed regardless of *dose*. The interaction between the two factors was not statistically significant [$F(27,279) = 1.37$, n.s.] (Fig 3, bottom panel).

## Discussion

The aim of this study was to determine whether the effect of imipramine to decrease sucrose ingestion could be observed even in absence of post-session administration.

In the first part of the experiment three testing sessions were performed 1-h after drug administration, i.e. at the time of the brain drug $C_{max}$ [19]. Imipramine treatment with 20 mg/kg resulted in reduced sucrose ingestion, indicated by the decrease of the lick number, which was observed in all sessions. This effect was accounted for by a reduction of burst number. A significantly reduced burst number was observed also with the doses of 5 and 10 mg/kg, but the consequences of these effects on overall ingestion were countered by a parallel increase in burst size, which was statistically significant only with the dose of 10 mg/kg (Fig 1, left panels). Even in the second part of the experiment–with the sessions (three plus three) being performed in an alternate order either 1-h or 24-h after drug treatment–imipramine 20 mg/kg resulted in reduced lick number. This effect was accounted for by reduced burst number. Imipramine at the lower doses did not affect lick number, burst number and burst size (Fig 1, right panels). These observations are consistent with the results of our previously published study [18]. Most importantly, the present results demonstrate that the reduction of the activation of licking behaviour can be induced regardless of imipramine post-session administration, since it was observed also in the sessions performed before the beginning of post-session administrations, i.e. in the sessions up to the the first "24-h after-treatment" session (day 12).

## DAY 1

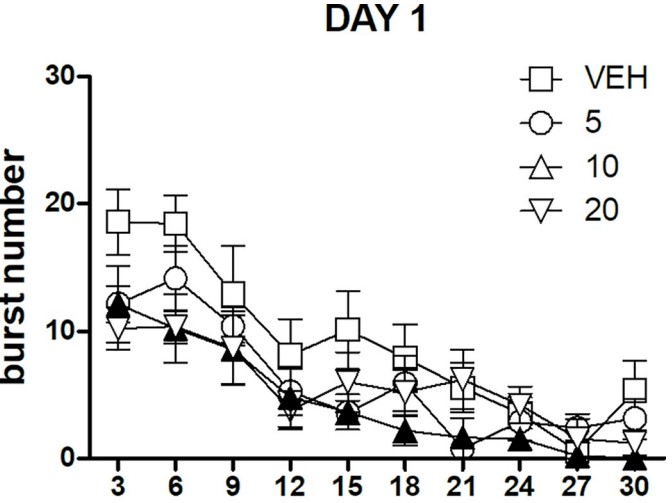

## DAY 5

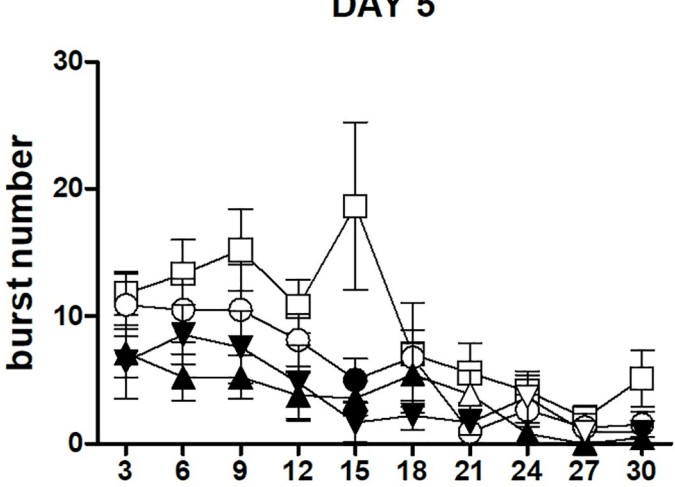

## DAY 8

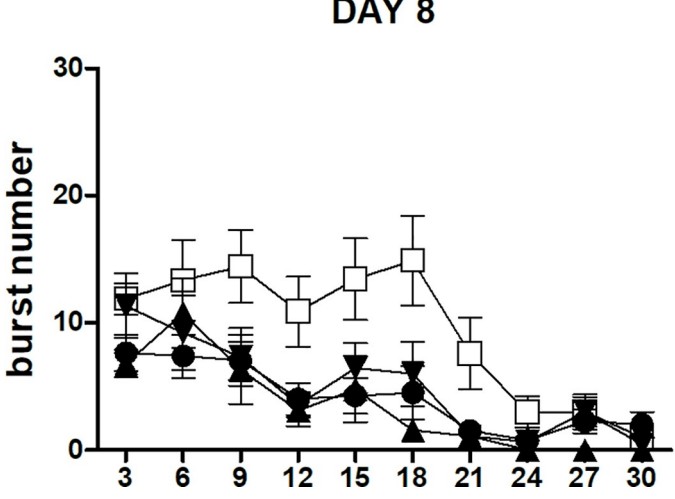

**Fig 3. Within-session time-course of burst number in the three sessions performed in the first part of the experiment.** Values represent the mean ± S.E.M. from 8–9 subjects. Black symbols indicate a statistically significant difference (P<0.05) with respect to the corresponding value of the vehicle-treated group (ANOVA followed by F-test for contrasts or Dunnett's test).

Thus, these observations, taken together with the observations of our previously published study, suggest that the reduced activation of licking behaviour (i) is due to reduced motivation and/or increased satiation, (ii) is likely related to imipramine effect at testing time, and (iii) does not require post-session administration. The effects of a treatment schedule involving only post-session administrations–without pre-session treatments–remain to be established.

Since burst size decrement in the course of the session might occur due to post-ingestional signals [10], in addition to whole-session data, we examined the data of the first 3-min time bin from each session, obtaining virtually superimposable results (S1 Fig). (See also S2 Fig, reporting the within-session time-course of burst size in the sessions performed in the first part of the experiment.) This observation is consistent with the results of previous studies reporting no change of burst size in the course of the session [4,5,18,22].

Consistently with the results of our previously published study [18], these effects were accompanied to reduced intra-burst lick rate (Fig 2, top panels), an effect which might suggest that the reduced ingestion might depend on motoric effects [27–29]. However, the effects of imipramine on burst number were more pronounced in the first part of the experiment, while the effects on the intra-burst lick rate were more pronounced in the second part of the experiment. The discrepancy in the time-course of these two measures suggests that the reduced activation of licking behaviour–indicated by reduced burst number–is not related to motor effects–indicated by reduced intra-burst lick rate. It might be worth noting that the intra-burst lick rate can be influenced also by experimental manipulations unlikely to be explained by motor impairment, such as changes in sucrose concentration [4,5,10], sodium depletion [30] and conditioned taste aversion [31]. In more detail, in the first part of the experiment only the dose of 20 mg/kg resulted in a reduction of the intra-burst lick rate, while in the second part of the experiment–but only in the sessions performed 1-h after drug treatment–also the dose of 10 mg/kg was effective. This observation is consistent with the hypothesis that imipramine effects on intra-burst lick rate depend on brain drug levels, which increase in the course of chronic treatment and are related to administration time with respect to testing sessions [19].

The increased latency to lick–observed with the dose of 20 mg/kg in the second and in the third session of the first part of the experiment (Fig 2, bottom left panel)–can be explained either as the consequence of motor impairment or as reduced motivation [4,22]. However, the observation that this effect is present only in the first part of the experiment–when the effects on burst number are more pronounced and the effects on intra-burst lick rate are less pronounced–support an interpretation in terms of reduced motivation. (But the absence of effect on this measure in the first session demands an explanation.)

In general, the design of this study does not allow to draw definitive conclusions as for the mechanisms underlying the differences between imipramine effects in the first and in the second part of the experiment, since they might be due either to the different treatment schedules–only pre-session administrations *versus* interposed pre-session and post-session administrations–or to the length of the imipramine treatment–which involves both a progressive increase of brain drug levels [19] and the possibility of adaptive changes [17,32,33]. Bearing in mind this limitation, the reduced effect of imipramine on burst number in the second part of the experiment might suggest that post-session administration does not result in a potentiation of this effect, as we previously observed with memantine administration [20,21].

The increased burst size observed with imipramine 10 mg/kg in the three sessions run in the first part of the experiment might suggest an increased hedonic response to sucrose (see Introduction). Since these sessions were performed before the beginning of post-session administrations, one might speculate that imipramine administration after licking tests might prevent the prohedonic effect of the successive imipramine pre-test administrations, exerting a post-session memory-related effect, such as conditioned taste-aversion (see [20,21]). This account might also help to explain the lack of this effect in our previously published study [18]. Further experiments–including a test assessing the ability of this imipramine administration schedule to induce conditioned taste-aversion–are necessary to test this hypothesis. However, in our previously published study [18], no increase in burst size was observed in the first two sessions, i.e. before the first post-session imipramine administration. As an alternative explanation, the possibility that the lack of effect of imipramine on burst size in the second part of the experiment might be the consequence of adaptive changes in response to the prolonged drug treatment [17,32,33] cannot be ruled out. Regardless of the mechanism involved, this effect does not appear to be a robust effect, in contrast to the reduced level of ingestion due to reduced burst number, which was consistently observed both here and in our previously published study [18]. It might be worth recalling that only a handful of studies in decades reported the ability of antidepressants to increase the sensitivity to rewards of different kinds in normal subjects [34–37], while there is a great deal of evidence showing the ability of antidepressants to reverse the reduced sensitivity to reward induced by stress, but without "prohedonic" effects in the control subjects [2,3].

Caution in regard to the interpretation of the observed increase of burst size as an increased hedonic response is advised by the analysis of the within-session time-course of burst number in the sessions showing an increased whole-session burst size (Fig 3). In a recent study, we have shown that the response to increased reward value–i.e. the response to increased hedonic impact–is not limited to an increase of whole-session burst size, but is characterised by an increase of burst number occurring at the beginning of the session [4]. A similar response pattern was observed also with clozapine, whose effect was interpreted as "prohedonic" by virtue of an increased whole-session burst size [38]. In contrast, the increased whole-session burst size observed in the present study was accompanied to reduced burst number, occurring since the beginning of the session. The association between increased whole-session burst size and reduced whole-session burst number, with reduced overall intake, was previously observed with the dopamine $D_1$-like receptor antagonist SCH 23390 in rats licking for water, and might represent a compensatory response to the reduced activation of ingestion [26,39]. Nonetheless, further experiments are necessary to determine whether the effect of imipramine on burst size might be accounted for by an increased hedonic impact.

These observations, taken together with the results of our previously published study [18], suggest that the primary effect of imipramine treatment was to reduce behavioural activation, due to reduced motivation and/or to increased satiation, with the possibility of an increased burst size as a compensatory response. The interpretation of imipramine effects as reduced motivation is in keeping with the finding that imipramine treatment reduced exploratory behaviour during habituation to a novel environment [40] and cocaine-seeking [41].

These results are consistent with the results of previous studies showing the ability of acute administration of antidepressant drugs in normal subjects to reduce the lick number for sweet solutions [13,14]. It might be worth noting that, at variance with the results from our lab and of Higgs and colleagues [14], the reduced lick number reported by Asin and colleagues [13] was accounted for by reduced burst size.

Consistently with the results of previous studies [18,42,43], treatment with imipramine suppressed body weight gain, with the doses of 10 and 20 mg/kg resulting in reduced average

body weight values with respect to the vehicle-treated group at the end of treatment (Table 2). This effect can be accounted for by the reduced activation of ingestive behaviour, regardless of whether it is due to reduced motivation or increased satiation. The burst size data, which show that this measure is either unmodified or increased (10 mg/kg imipramine, first part of the experiment), do not lend support to the possibility that the reduced ingestion might be due to a drug-induced state of malaise, which is characterised by a reduction of this measure (e.g. [31,44]).

In conclusion, in keeping with the results of a previously published study [18], treatment with imipramine in normal rats–i.e. not subjected to manipulations aimed to model aspects of depression–yields a reduction of sucrose ingestion accounted for by reduced burst number, possibly due to reduced motivation and/or to a potentiation of post-ingestional satiation signals. Thus, these results might be relevant in the explanation of the effects of imipramine in models of drug-seeking and in body weight gain reduction in rats, but not in accounting for the antidepressant therapeutic effect. Most importantly in relation to the aim of this study, these results demonstrate that the reduced behavioural activation induced by imipramine does not require post-session drug administration. The increased burst size observed in the first half of the treatment period–which was not observed in our previous study–might be interpreted as a prohedonic effect and/or as a compensatory effect.

## Supporting information

**S1 Data. All individual data used for statistical analysis and figures.**
(XLS)

**S1 Fig. Early session burst size.**
(TIF)

**S2 Fig. Within-session time-course of burst size in the first part of the experiment.**
(TIF)

## Author Contributions

**Conceptualization:** Paolo S. D'Aquila, Adriana Galistu.

**Data curation:** Adriana Galistu.

**Formal analysis:** Paolo S. D'Aquila, Adriana Galistu.

**Funding acquisition:** Paolo S. D'Aquila.

**Investigation:** Paolo S. D'Aquila, Adriana Galistu.

**Supervision:** Paolo S. D'Aquila.

**Writing – original draft:** Paolo S. D'Aquila, Adriana Galistu.

**Writing – review & editing:** Paolo S. D'Aquila.

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
