## [Decision Letter · Decision Letter 0]

13 Nov 2020

PONE-D-20-29592

Further characterization of the effect of the prototypical antidepressant imipramine on the microstructure of licking for sucrose

PLOS ONE

Dear Dr. D'Aquila,

Thank you for submitting your manuscript to PLOS ONE. After careful consideration, we feel that it has merit but does not fully meet PLOS ONE’s publication criteria as it currently stands. Therefore, we invite you to submit a revised version of the manuscript that addresses the points raised during the review process.

Both reviewers recognized strengths of your study and the experimental design. However, there are a number of areas in which they felt additional analysis and commentary would help to understand the effects of imipramine. Please address each of those points in your revised manuscript. In addition, it is essential that all data (e.g. individual data points that underlie averages) are supplied or deposited online to comply with journal policy.

We look forward to receiving your revised manuscript.

Kind regards,

James Edgar McCutcheon, Ph.D.

Academic Editor

PLOS ONE

Journal Requirements:

2. We note that there is quite a bit of text overlap between your paper and the following papers from your author group:

https://doi.org/10.1016/j.physbeh.2020.113032

https://doi.org/10.1016/j.ejphar.2020.173468

We would like to make you aware that copying extracts from previous publications, especially outside the methods section, word-for-word is unacceptable - whether the work is your own or another author group. In addition, the reproduction of text from published reports has implications for the copyright that may apply to the publications.

Please revise the manuscript to rephrase the duplicated text and provide details as to how the current manuscript advances on previous work .We will carefully review your manuscript upon resubmission, so please ensure that your revision is thorough. Thank you for your attention to this matter!

Reviewers' comments:

Reviewer's Responses to Questions

**Comments to the Author**

1. Is the manuscript technically sound, and do the data support the conclusions?

Reviewer #1: Yes

Reviewer #2: Partly

2. Has the statistical analysis been performed appropriately and rigorously? 

Reviewer #1: Yes

Reviewer #2: No

3. Have the authors made all data underlying the findings in their manuscript fully available?

Reviewer #1: Yes

Reviewer #2: No

4. Is the manuscript presented in an intelligible fashion and written in standard English?

Reviewer #1: Yes

Reviewer #2: Yes

5. Review Comments to the Author

Reviewer #1: Overall, the experiment is very well designed and the analysis of the dose-dependant effect and the within session time course of imipramine’s effect brings important information about antidepressant effects on ingestive behaviour. I only have few comments:

1) The choice of either F-tests or Newman-Keuls post hoc tests after significant interactions was unclear both in the Methods and Results section. What informed these decisions?

2) The analysis of within session effect is especially interesting. I was wondering why the two first sessions have been included in this analysis (instead of only the 1st one or all of them)?

3) The analysis of the whole session number of licks per burst showed a clear significant increase during the first sessions in animals treated with 10 mg/kg imipramine. Even if it was not significant, a similar pattern can be noted with 5 and 20 mg/kg. I was wondering if the number of licks per burst was constant during the session or was evolving with the ingestion of sucrose under imipramine.

4) High doses of imipramine induced a decrease in body weight compared to vehicle injected rats but also compared to the rats’ original weight. Did the authors notice a decrease in food consumption beyond the sucrose ingestion procedure?

5) Animals receiving the highest dose of imipramine (20 mg/kg) seem to have an increased latency to lick associated with a general decrease of all the other licking parameters including the intra-burst lick rate. As suggested by the authors, this could reflect a potential locomotor effect. Even if it is beyond the scope of the present paper, I was wondering if the authors measured locomotor activity in imipramine-treated animals?

6) Colour code of Figures is sometimes a bit confusing with some symbols appearing white in some panels and black in others. This should be harmonized.

7) Authors should add a sentence about data availability and where data can be publicly found (either as Supplemental file or on a data repository) to comply with the journal's guideline.

Reviewer #2: In this study, D’Aquila and Galistu examined the effects of repeated acute (1 hour prior) and delayed (24 hour prior) systemic administration of imipramine on sucrose intake and licking microstructure in rats. Analyses of licking patterns revealed that mid-high doses of this drug administered just before access to sucrose led to reductions in total licks over 30 minutes. This outcome was mainly accomplished via a reduction in the number of licking bursts and slower lick rate in response to imipramine, which is interpreted as an amplification of satiation processes (or reduction in reward processes). A similar intake pattern was observed if the drug was administered 24 hours prior to sucrose access at least at the highest dose, though this was less robust. The results are taken to suggest that imipramine’s acute effects are sufficient to reduce sucrose intake. This expands upon this group’s previous work, addressing a caveat from their earlier cited study whose design did not permit differentiation between the direct and indirect (e.g., CTA learning) effects of the drug on sucrose consumption. Although the study expands upon the prior work in an incremental fashion, there are several outstanding interpretational and technical concerns. These are listed below.

First, the overall design and rationale for the design are somewhat difficult to follow as presented. A timeline would be helpful in this regard. For example, the rats were acclimated to the sucrose in the lickometers, but for how long, how many sessions, with food or water deprivation? During the interposed non-test sessions, were the rats provided sucrose as usual?

If the primary outcome of interest here was burst number and total intake, it’s unclear why the groups were matched on burst size.

The rationale for selecting a 400 ms pause criterion is not clear. Previous work in rats found that a >1 second pause criterion captured both short and long breaks between active licking bursts. Perhaps this criterion would decrease some of the noise in the burst data. Spector AC, Klumpp PA, Kaplan JM. Analytical issues in the evaluation of food deprivation and sucrose concentration effects on the microstructure of licking behavior in the rat. Behav Neurosci. 1998 Jun;112(3):678-94.

With respect to the prior study, upon which this one is based, if CTA was one possible mechanism, then this would be expected to appear as an increase in burst number, with a corresponding decrease in burst size, as rats would less readily consume the substance. This would reflect a devaluation of the orosensory properties of the sucrose. Moreover, one would expect, based on the previous literature, that the initial lick rate (or first burst size), which are rendered by the orosensory input of the stimulus, prior to the onset of any postingestive effects, would be reduced in the group that received the drug from the first exposure to the last. This too would reflect a learned or indirect effect of the drug or a primary effect on reward versus satiation.

Familiarization with sucrose prior to testing with the drug would be expected to attenuate any learning about the stimulus here. For this reason, it would be helpful to clarify the nature of the acclimation phase and acknowledge that as a caveat in the present design.

Analyses of the interlick interval distributions would provide a more reliable check on motor impairments produced by the drug.

Nevertheless, there are potential indications in the data presented here for unconditioned and conditioned/repeated effects of the drugs. The effects of the drug appear weaker on the initial session (day 1) relative to subsequent sessions (including in latency to initiate licking). There also appears to be substantial weight loss from start to finish in the high dose groups. For this reason, it might be worthwhile to analyze /discuss the initial session (day 1) as compared to the subsequent 1-hour prior sessions. Do total licks and licking patterns return to normal in between the tests (the “off” days)? Such data could be informative with respect to teasing apart the effects modulated by drug state contexts/conditioning and unconditioned effects of the drug. Given the strong conclusions offered in lines 262-266 and in the final paragraph of the discussion that reduced sugar consumption “does not depend on post-session administration” and is related to the acute effect of the drug on board during testing, these issues need to be resolved more fully. In general, the repeated drug exposure nature design and order effects are not sufficiently considered. A separate experiment specifically assessing whether these doses condition avoidance of a novel flavor would help resolve these issues.

There is certainly a trend for higher doses of imipramine to reduce burst number and intra-burst lick rate in the 24-hour post injection condition- in some cases this reaches significance. Therefore, a fuller discussion of potential long term effects of the drug are warranted, while keeping in mind that these rats had extensive exposure to this drug prior to the 24 hour post-injection phase.

The initial phase (1 hr) should be analyzed separately from the later phase (interposed with 24 hr) sessions.

Discussion of the relationship between burst size and number across sessions 1 and 5 in lines 296-301 is unclear, as only burst numbers are plotted. The argument appears to be that reduced burst number and increase burst size typically results in a similar total licks outcome, not a reduced lick outcome as seen in this study. Parallel plots of burst size (or licks/bin) across these bins would perhaps help make the point clearer.

The argument in Discussion lines 319-324 is not entirely convincing, given burst size is quite variable, and perhaps increased early on in the 10 mg dose condition. Moreover, it would be worth considering whether this “early” increase in burst size in the 10 mg condition that disappears after the first of the 24 hour post-injection sessions is related to those treatments or just a function of the repeated exposure.

Decreased burst number is referred to as satiety throughout (e.g., line 310), when this perhaps more accurately tracks satiation in these short term sessions.

6. PLOS authors have the option to publish the peer review history of their article (what does this mean?). If published, this will include your full peer review and any attached files.

Reviewer #1: **Yes: **Fabien Naneix

Reviewer #2: No

---

## [Author Response · Author response to Decision Letter 0]

10 Dec 2020

Response to Reviewers, including the response to Journal and Editor comments

The text by the Editor, the Journal and the Reviewers is in Italics. Please notice that in the Revised Manuscript with Track Changes the new text is highlighted with a light grey background. In square brackets are indicated citations from the Reference list of the submitted paper.

Editor

Both reviewers recognized strengths of your study and the experimental design. However, there are a number of areas in which they felt additional analysis and commentary would help to understand the effects of imipramine. Please address each of those points in your revised manuscript. In addition, it is essential that all data (e.g. individual data points that underlie averages) are supplied or deposited online to comply with journal policy.

Response: The paper was revised according to the Reviewers' comments and the data were reanalysed. All individual data were provided (Supporting Information File, S1 Data). Below our detailed responses to the Reviewers' comments and to the Journal's Editorial Office.

Journal

Response: The article text was formatted according to the style requirements of PLoS One. Files were named according to the instructions of the Journal.

2. We note that there is quite a bit of text overlap between your paper and the following papers from your author group:

https://doi.org/10.1016/j.physbeh.2020.113032

https://doi.org/10.1016/j.ejphar.2020.173468

[…]. Please revise the manuscript to rephrase the duplicated text and provide details as to how the current manuscript advances on previous work. […]

Response: The text overlapping with the above linked publications was rephrased – the line numbers given below refer to the Revised Manuscript with Track Changes. Please see lines 42-44, 46-57, 60-68, 69-74, 399-402, 474-476, 493-495.

 Please notice that several standard expressions were left unchanged in the revised text: Abstract: “[...] induced a dose-dependent reduction of the ingestion of a 10% sucrose solution […] reduction of the licking burst number [...]” (lines 17-18); “[...] the experimental sessions were performed in an alternate order [...] either 1-h or 24-h after imipramine administration.” (lines 19-20); Introduction: “[...] of psychological constructs relevant to depression, such as pleasure – hedonic impact/reward evaluation –, and motivation – incentive salience attribution/behavioural activation.” (lines 44-46); “[...] ten experimental sessions were performed in an alternate order […] either 1-h or 24-h after imipramine administration [...]” (lines 75-76).

Reviewers' comments

Reviewer #1: Overall, the experiment is very well designed and the analysis of the dose-dependant effect and the within session time course of imipramine’s effect brings important information about antidepressant effects on ingestive behaviour. I only have few comments:

We wish to thank the Reviewer for the appreciation of our work.

1) The choice of either F-tests or Newman-Keuls post hoc tests after significant interactions was unclear both in the Methods and Results section. What informed these decisions?

 In the submitted revision, the comparisons based on the main effect of the factor “dose”, i.e. the comparisons between each of the three imipramine dose groups and the vehicle-treated group, were analysed with the Dunnett's test, which is considered an appropriate post-hoc test for multiple comparisons versus a common control group (Upton G. & Cook I., A Dictionary of Statistics. 2nd edition, Oxford University Press 2006).

 The use of F-tests (planned comparisons or “a priori” tests) in presence of a significant interaction between factors is justified when just a set of all the possible comparisons are of interest. Indeed, with “a priori” tests – as opposed to the case of post-hoc tests – only the comparisons between levels within the same factor or between different factors at the same level are allowed (Winer B. J., Statistical principles in experimental design. McGraw Hill; 1971).

 To deal with this comment the citations above have been included in the revised manuscript.

2) The analysis of within session effect is especially interesting. I was wondering why the two first sessions have been included in this analysis (instead of only the 1st one or all of them)?

Response: Prompted by this remark, we included in the revised version the analysis and the graphs of all the sessions showing a whole-session burst size increase (Fig 3). In the new data analysis (as suggested by Reviewer #2 point 8), the sessions showing an increased whole-session burst size turned out to be only the three sessions run in the first part of the experiment (please see Fig 1, bottom panels).

3) The analysis of the whole session number of licks per burst showed a clear significant increase during the first sessions in animals treated with 10 mg/kg imipramine. Even if it was not significant, a similar pattern can be noted with 5 and 20 mg/kg. I was wondering if the number of licks per burst was constant during the session or was evolving with the ingestion of sucrose under imipramine.

Response: This is an important issue, but the literature in this regard is not consistent, with some studies reporting a decline of burst size within a session and other reporting no change. To deal with the Reviewer's comment, we performed an additional analysis of the early-session burst size data (data from the 1st 3-min time bin). The results (lines 287-294 of the Revised Manuscript with Track Changes, or lines 229-236 of the clean Manuscript, S1 Fig) were virtually superimposable to the results of the whole-session data (Fig 1, bottom panels). Citations of previous reports and discussion of the present results were dealt with in the Discussion (lines 394-398 Revised Manuscript with Track Changes or lines 321-325 clean Manuscript).

4) High doses of imipramine induced a decrease in body weight compared to vehicle injected rats but also compared to the rats’ original weight. Did the authors notice a decrease in food consumption beyond the sucrose ingestion procedure?

Unfortunately the food consumption was not recorded. Indeed, as noted by the Reviewer, imipramine 20 mg/kg resulted in a net weight loss. In the revised version, body weight data analysis was integrated with the comparisons between start and end of treatment (Revised Manuscript with Track Changes: Results lines 188-190, Table 2 page 9, and Discussion lines 483-485)

5) Animals receiving the highest dose of imipramine (20 mg/kg) seem to have an increased latency to lick associated with a general decrease of all the other licking parameters including the intra-burst lick rate. As suggested by the authors, this could reflect a potential locomotor effect. Even if it is beyond the scope of the present paper, I was wondering if the authors measured locomotor activity in imipramine-treated animals?

In the paper we argue that while reduced ingestion can be accounted for either by reduced motivation/increased satiation or by impaired motor function, the comparative inspection of the whole-experiment time-course of latency, burst number, and intra-burst lick rate suggests that motor impairment does not give a satisfactory account of the results (lines 399-423 Revised Manuscript with Track Changes or lines 326-347 clean Manuscript). We did not measure locomotor activity in these subjects, but we have earlier data showing the ability of imipramine to reduce locomotor activity during habituation to a novel environment (please see reference 41). However, even data on locomotor activity, in principle, can be accounted for either by motivational or by motor mechanisms.

6) Colour code of Figures is sometimes a bit confusing with some symbols appearing white in some panels and black in others. This should be harmonized.

As indicated in the figure legends, the black colour indicates a significant difference with respect to the vehicle-treated group in the corresponding time bin. This explains the apparent inconsistency between panels.

7) Authors should add a sentence about data availability and where data can be publicly found (either as Supplemental file or on a data repository) to comply with the journal's guideline.

The data are now provided as Supporting Information (S1 Data).

Reviewer #2:

In this study, D’Aquila and Galistu examined the effects of repeated acute (1 hour prior) and delayed (24 hour prior) systemic administration of imipramine on sucrose intake and licking microstructure in rats. Analyses of licking patterns revealed that mid-high doses of this drug administered just before access to sucrose led to reductions in total licks over 30 minutes. This outcome was mainly accomplished via a reduction in the number of licking bursts and slower lick rate in response to imipramine, which is interpreted as an amplification of satiation processes (or reduction in reward processes). A similar intake pattern was observed if the drug was administered 24 hours prior to sucrose access at least at the highest dose, though this was less robust. The results are taken to suggest that imipramine’s acute effects are sufficient to reduce sucrose intake. This expands upon this group’s previous work, addressing a caveat from their earlier cited study whose design did not permit differentiation between the direct and indirect (e.g., CTA learning) effects of the drug on sucrose consumption. Although the study expands upon the prior work in an incremental fashion, there are several outstanding interpretational and technical concerns. These are listed below.

1) First, the overall design and rationale for the design are somewhat difficult to follow as presented. A timeline would be helpful in this regard. For example, the rats were acclimated to the sucrose in the lickometers, but for how long, how many sessions, with food or water deprivation? During the interposed non-test sessions, were the rats provided sucrose as usual?

Response: A timeline of the experiment was included in the revised version (Table 1, page 5 Manuscript with Track Changes, page 4 clean Manuscript). Moreover, we integrated the “Procedures” paragraph with the information requested by the Reviewer. In the “Procedures” section – please see Manuscript with Track Changes – it is reported the number (9 in 15 days, lines 147-148) and duration (30-min, line 144) of training session, i.e. of the acclimation to sucrose. It was added a statement that the animals were not food/water deprived and that sucrose was made available to the animals only during training/test sessions (lines 148-149). We hope that these changes resulted in improved clarity. In particular, it should be clear that there are not any interposed non-test sessions, but only treatment days with no experimental sessions – with no sugar available to the subjects in non-test days, as now explicitly stated.

2) If the primary outcome of interest here was burst number and total intake, it’s unclear why the groups were matched on burst size.

Response: We followed the procedure usually adopted in our lab, including our previous study on which this study builds upon [18], when we had no expectations as for which measure should be considered as the primary outcome. Perhaps, one might surmise that the whole-session burst size as a matching measure might be the best choice because it is calculated from both lick number (i.e. the measure of intake) and burst number (whole-session burst size = whole session lick number / whole-session burst number). However, we have no data to support this choice, other than the internal consistency of the data obtained in our lab and their consistency with the results from other labs. Nonetheless, in order to deal with this observation, we included in the data file provided in the Supporting Information section (S1 Data, page 1) also the lick number, burst number and burst size data of the matching day, along with the results of the ANOVA for each set of data. No statistically significant differences between the groups were found for any of these measures.

3) The rationale for selecting a 400 ms pause criterion is not clear. Previous work in rats found that a >1 second pause criterion captured both short and long breaks between active licking bursts. Perhaps this criterion would decrease some of the noise in the burst data. Spector AC, Klumpp PA, Kaplan JM. Analytical issues in the evaluation of food deprivation and sucrose concentration effects on the microstructure of licking behavior in the rat. Behav Neurosci. 1998 Jun;112(3):678-94.

Response: The simple reason for choosing such a pause criterion was for consistency with our previous study [18]. Nonetheless, this response begs the question of the reason of this choice in our previous studies in the first place. To answer this question, it is necessary a detailed response.

 Davis & Smith (1992) – reference n. 5 of the submitted paper – have shown that licking consists of 

bursts of licks with interlick intervals (ILIs) of approximately 150 ms. These bursts of licks are separated by two types of pauses, one of about 300 ms and the other >500. Based on these observations they identify three regions of distributions: “(a) the within-burst distribution (ILIs ≤ 250 ms), (b) the inter-burst interval (IBI) distribution (ILIs >250 and ≤ 500 ms), and (c) the intercluster interval (ICI) distribution (ILI > 500 ms).”

 Based on these distributions, Davis and Smith define the bursts of licking as “runs of licks” with ILIs ≤ 250 ms. According to Davis and Smith “[...]sustained periods of rhythmic tongue activity can be seen as being organized into runs of bursts of rhythmic tongue movements, with each burst separated by an interval corresponding roughly to one period of the tongue extension-retraction cycle. These runs of bursts form cluster of bursts. A cluster of bursts, therefore consists of a run of ILIs less than or equal to 500 ms. […] Defining a cluster in this way provides a measure of the time that the rat is licking, with interruptions lasting for only a single cycle of the extension-retraction cycle of the tongue.”

 Thus, Davis and Smith define the “clusters of bursts” or simply “clusters” – which they consider as bouts of ingestion – as runs of licks with ILIs ≤ 500 ms, which they call IBIs (inter-burst intervals).

 In our study, a burst of licking is defined by a pause criterion of >400 milliseconds. We chose this criterion 10 years ago (D'Aquila, 2010 [22]), following the research group of Steve Cooper and Susan Higgs. They adopted such criterion because it had been established in previous studies that an interval of 400 ms was just longer than the break point in a log survivor plot of ILIs (see Higgs and Cooper 1998, reference 23 of the revised version). This piece of information was added to the Method section, subsection “Apparatus and microstructural measures” (lines 135-136 Revised Manuscript with Track Changes, 119-120 clean Manuscript).

 Importantly, this value (>400 ms) is more than twice the average within-burst inter-lick interval. This is important because IBIs are about 300 ms in duration, i.e. twice the average ILI. Indeed this value is close to the “intercluster interval” separating lick clusters, that Davis and Smith set at >500 ms. Thus, licking bursts defined according to this pause criterion (>400 ms) share with clusters – as defined by Davis and Smith – an important feature: they are series of “bursts” – according to Davis and Smith's definition – separated by one period of the cycle of extension-retraction of the tongue. Please notice that the number of IBIs >400 ms <500 ms– according to the distribution frequencies reported in Davis and Smith 1992 – appears to be negligible. In this regard, see Fig. 3 in Davis and Smith 1992 [5], showing the distribution of ILIs and IBIs in a single subject (rat): The mean and standard deviation of the distribution of frequency for ILIs was 136 ± 14 ms with a peak frequency of about 990, while for IBIs was 279 ± 33 ms – i.e. twice the value of the within-burst ILI distribution – with a peak frequency of 48 (i.e. about a 20th). Therefore, the definition of burst adopted in the present study (and in all our previous studies and in almost all the studies performed in rats by the research groups of Steve Cooper and Susan Higgs) is functionally similar to the cluster as defined by Davis and Smith (1992).

 It is also important to stress that the vast majority of ILIs included in a burst defined with the pause criterion of 400 milliseconds – i.e. the pause criterion adopted in the submitted study – should fall in the distribution region of ILIs ≤ 250 milliseconds. The few longer inter-lick intervals (called IBIs by Davis and Smith) falling in the distribution region >250 ≤ 400 milliseconds, can be considered – as well as the ILIs ≤ 250 – related to the activity of the central pattern generator.

 Please also notice that Spector et al. 1998 [10] compare the 1 sec pause criterion with a pause criterion of 300 milliseconds (and with pause criteria > 1 sec, but this is not relevant in this argumentation) and describe the results obtained with the 1 sec pause criterion as similar to the results obtained by Davis and Smith with the pause criterion of 0.5 sec. In keeping with these considerations, Higgs et al. 2003 (Psychopharmacology 165:370-377), in a study using the 0.4-sec criterion, repeated the analysis defining licking bursts with a pause criterion of 1-sec, observing that in most cases, the data were similar regardless of which criterion was adopted.

4) With respect to the prior study, upon which this one is based, if CTA was one possible mechanism, then this would be expected to appear as an increase in burst number, with a corresponding decrease in burst size, as rats would less readily consume the substance. This would reflect a devaluation of the orosensory properties of the sucrose. Moreover, one would expect, based on the previous literature, that the initial lick rate (or first burst size), which are rendered by the orosensory input of the stimulus, prior to the onset of any postingestive effects, would be reduced in the group that received the drug from the first exposure to the last. This too would reflect a learned or indirect effect of the drug or a primary effect on reward versus satiation.

Familiarization with sucrose prior to testing with the drug would be expected to attenuate any learning about the stimulus here. For this reason, it would be helpful to clarify the nature of the acclimation phase and acknowledge that as a caveat in the present design.

Response: (i) CTA was suggested as a possible mechanism to account for the observation that memantine post-session administration (reference 20) resulted in a dramatic decrease of sucrose intake (with levels of ingestion close to naught). However, whether or not that effect could be explained as CTA remains to be established. (ii) In a recent study, we observed that imipramine administered in an alternate order either before or after testing sessions, resulted in a decrease of sucrose intake. (iii) Building upon the results of these two studies, and regardless of the interpretation of the memantine experiment data in terms of CTA, the aim of the present study was “to determine whether imipramine-induced decrease of sucrose ingestion could be observed even in absence of post-session administration” (Revised Manuscript with Track changes: lines 90-91, unchanged from the 1st submission). The results showed that imipramine reduced sucrose ingestion since the beginning of the experiment, well before commencing post-session administrations. This experiment was not designed to sort out the problem of the interpretation of post-session administration effects in terms of CTA. While we think that the results of the first part of the experiment provide a clear and unequivocal response to the question posed by the objective of this study, we think that the Reviewer's concerns do apply to the interpretation of the results of the second part of the experiment. Thus, to deal with these observations, we made changes to the text aimed at clarifying these issues. In particular (see Revised Manuscript with Track Changes):

Introduction:

- “As for the interpretation of this observation [i.e. the effect of memantine post-session administration], we suggested the possibility of the development of a memory-related effect, such as conditioned taste-aversion [20,21]. However, this question remains to be settled by further experiments.” (lines 87-89, see also lines 82-83 with the cancelled text).

Discussion:

- “The effects of a treatment schedule involving only post-session administrations – without pre-session treatments – remain to be established.” (lines 392-393)

- See also the paragraph at page 18, lines 424-429, dealing with the differences between the two parts of the experiment.

- “Further experiments – including a test assessing the ability of this imipramine administration schedule to induce conditioned taste-aversion – are necessary to test this hypothesis.” (lines 443-444).

- Please also notice that the conclusive remarks (last paragraph of the Discussion), are based solely on the results of the first part of the experiment.

5) Analyses of the interlick interval distributions would provide a more reliable check on motor impairments produced by the drug.

Response: The measure of the licking microstructure which is most commonly used as an index of motor competence is the within-burst lick rate or, alternatively, the average ILI (i.e. the interlick interval corresponding to the distribution region below the interval criterion adopted to define licking bursts), which is the reciprocal of the within-burst lick rate (e.g. Spector et al. 1998, reference 10; Lydall et al., 2010, Psychopharmacology 209, 153-62). It is true that also the ILI distribution was used for this purpose. But only in that it allows to infer the within-burst lick rate: “[...] Note that the equivalence of these interlick distributions under these three test conditions indicated no alteration of the mean rate of licking [...]” (Schneider et al., 1990, Eur J Pharmacol 186:61-70, Figure 3 legend, from the research group of Gerard Smith and John Davis). The figure shows the interlick interval distribution of three experimental conditions in the region between 100 and 200 licks/sec. A number of labs measure the within-burst lick rate (or the average ILI) using pause criteria above the value of the within-burst ILI (which is ≤ 250 ms), and which correspond to the different pause criteria used to define the licking burst/cluster/bout – depending on the terminology adopted (for details see response to point 3 above). This is justified by the fact that the vast majority of ILI below 1 sec fall in the distribution region of the ILIs according to the definition of Davis and Smith (see Spector et al., 1998, and the response to point 3 above). Below some examples:

0.4 sec: Higgs and Cooper, 1998 [23], 2000 (Eur J Pharmacol, 409:73-80); our own lab [4,18,20-22,38,39]

0.5 sec: Dwyer's lab, e.g. Lydall et al., 2010 (Psychopharmacology 209, 153-62)

1 sec: Baird et al. 2011 (Am J Physiol Regul Integr Comp Physiol 301: R1044–R1056).

Higgs and Cooper 1998, along with the intra-bout lick rate, reported also the ILI distribution frequency, with the two sets of measures providing consistent results.

6) Nevertheless, there are potential indications in the data presented here for unconditioned and conditioned/repeated effects of the drugs. The effects of the drug appear weaker on the initial session (day 1) relative to subsequent sessions (including in latency to initiate licking). There also appears to be substantial weight loss from start to finish in the high dose groups. For this reason, it might be worthwhile to analyze /discuss the initial session (day 1) as compared to the subsequent 1-hour prior sessions. Do total licks and licking patterns return to normal in between the tests (the “off” days)? Such data could be informative with respect to teasing apart the effects modulated by drug state contexts/conditioning and unconditioned effects of the drug. Given the strong conclusions offered in lines 262-266 and in the final paragraph of the discussion that reduced sugar consumption “does not depend on postsession administration” and is related to the acute effect of the drug on board during testing, these issues need to be resolved more fully. In general, the repeated drug exposure nature design and order effects are not sufficiently considered. A separate experiment specifically assessing whether these doses condition avoidance of a novel flavor would help resolve these issues.

Response: (Line numbers refer to the Revised Manuscript with Track Changes) To deal with these remarks, in the revised version it is acknowledged that some of the observed effects – especially the difference between the first and the second part of the experiment, might be due to different mechanisms, including the effect of repeated drug exposure (lines 413-416, 424-429, 448-450). As for the data from the “off” days, please see above the response to point 1. We hope that the integrated “Procedures” paragraph was successful in making it clear that there are no data from the “off” days, because they are simply days in which the animals received drug treatment but no measurement was performed. The effect in the first part of the experiment (following the new analysis suggested by this Reviewer, see below) – i.e. the effects which are relevant for the primary outcome of this experiment – do not show variations with respect to the first session, with the exception of the latency values (Latency new discussion in lines 417-423)

 Nonetheless, as rightly pointed out by the reviewer, to settle the questions raised in these comments, especially in regard of the results of the second part of the experiment, specific experiments are required. This is acknowledged in the revised text (please see Reply to point 4 above).

7) There is certainly a trend for higher doses of imipramine to reduce burst number and intra-burst lick rate in the 24-hour post injection condition- in some cases this reaches significance. Therefore, a fuller discussion of potential long term effects of the drug are warranted, while keeping in mind that these rats had extensive exposure to this drug prior to the 24 hour post-injection phase.

Response: (Line numbers refer to the Revised Manuscript with Track Changes) In the revised version it is discussed the issue of the potential long term effects of imipramine (lines 424-429). See also response to the previous point. The results of the new analysis do indeed show that the effects on intra-burst lick rate are stronger in the second part of the experiment, but more pronounced in the sessions performed 1-h after treatment (See lines 410-416 for a Discussion).

8) The initial phase (1 hr) should be analyzed separately from the later phase (interposed with 24 hr) sessions.

Response: We are particularly grateful to the Reviewer for this suggestion. Indeed, a separate analysis of the first part allows to separate the first part of the experiment – leading to unequivocal conclusions in relation to the stated aim of the experiment – from the second part of the experiment, which should be considered as “exploratory”, and whose interpretation requires further investigations.

9) Discussion of the relationship between burst size and number across sessions 1 and 5 in lines 296-301 is unclear, as only burst numbers are plotted. The argument appears to be that reduced burst number and increase burst size typically results in a similar total licks outcome, not a reduced lick outcome as seen in this study. Parallel plots of burst size (or licks/bin) across these bins would perhaps help make the point clearer.

Response: We omitted to specify “whole-session” burst size. Indeed, the previous observations on the within-session response pattern of the burst number are related to whole-session burst size, and not to the within-session time-course of this measure. In particular, we have shown that the response to an increase of the reward value is characterized not only by an increased whole-session burst size but also by an increased burst number at the beginning of the session. Moreover, the present results show that burst size did not change within the session (see response to point 3 of Reviewer 1). In the revised version we integrated the information adding “whole-session” to the text where appropriate (see lines 460-467 Revised Manuscript with Track Changes).

10) The argument in Discussion lines 319-324 is not entirely convincing, given burst size is quite variable, and perhaps increased early on in the 10 mg dose condition. Moreover, it would be worth considering whether this “early” increase in burst size in the 10 mg condition that disappears after the first of the 24 hour post-injection sessions is related to those treatments or just a function of the repeated exposure.

Response: (Line numbers refer to the Revised Manuscript with Track Changes) The text was reformulated taking into account this comment: lines 486-490. The possibility that the loss of effect of imipramine 10 mg/kg in the second part of the experiment was just a function of the repeated drug exposure was considered in the revised version (lines 448-450).

11) Decreased burst number is referred to as satiety throughout (e.g., line 310), when this perhaps more accurately tracks satiation in these short term sessions.

Response: “Satiety” was changed into “satiation” throughout the manuscript.

***

Finally, we wish to thank both the Editor and the Reviewers for their helpful and constructive comments, which – we believe – allowed us to improve the quality of the submitted manuscript.

---

## [Decision Letter · Decision Letter 1]

23 Dec 2020

PONE-D-20-29592R1

Further characterization of the effect of the prototypical antidepressant imipramine on the microstructure of licking for sucrose

PLOS ONE

Dear Dr. D'Aquila,

Thank you for submitting your manuscript to PLOS ONE. After careful consideration, we feel that it has merit but does not fully meet PLOS ONE’s publication criteria as it currently stands. Therefore, we invite you to submit a revised version of the manuscript that addresses the points raised during the review process.

Both reviewers recognized that substantial improvement had been made to the paper since the first submission. However, before acceptance once reviewer has still requested some additional details and presentation of a little more of the within-session data. Please attend to their comments before re-submitting.

We look forward to receiving your revised manuscript.

Kind regards,

James Edgar McCutcheon, Ph.D.

Academic Editor

PLOS ONE

Reviewers' comments:

Reviewer's Responses to Questions

**Comments to the Author**

1. If the authors have adequately addressed your comments raised in a previous round of review and you feel that this manuscript is now acceptable for publication, you may indicate that here to bypass the “Comments to the Author” section, enter your conflict of interest statement in the “Confidential to Editor” section, and submit your "Accept" recommendation.

Reviewer #1: All comments have been addressed

Reviewer #2: (No Response)

2. Is the manuscript technically sound, and do the data support the conclusions?

Reviewer #1: Yes

Reviewer #2: Partly

3. Has the statistical analysis been performed appropriately and rigorously? 

Reviewer #1: Yes

Reviewer #2: Yes

4. Have the authors made all data underlying the findings in their manuscript fully available?

Reviewer #1: Yes

Reviewer #2: Yes

5. Is the manuscript presented in an intelligible fashion and written in standard English?

Reviewer #1: Yes

Reviewer #2: Yes

6. Review Comments to the Author

Reviewer #1: (No Response)

Reviewer #2: The paper is greatly improved by reframing of it in terms of examining the “immediate” effects of the imipramine on sucrose licking patterns in the absence of interposed post-session administrations, as opposed to invoking some examination of learning or memory. In general, the discussion is a lot more cautious about mechanism, which is appropriate given the design and data. I still have a couple of outstanding concerns.

Perhaps the authors could add some indication of which sessions are included in part 1 and 2 in the table. But, in general, the table helps clarify the protocol.

I would like to see more of the within session data (plotted across time). Per my previous review, I thought it would be important to examine burst size in addition to be able to interpret the burst number data across the session. The authors now present the first 3 min burst size in the supporting information, but not beyond that. The authors appear to favor the interpretation that imipramine reduces behavioral motivation or increases satiation. But for the 10 mg dose (and maybe the 5), the data could just as easily be interpreted as increase in “hedonic value” based on the framework provided by the authors in the introduction. In fact, the first 3 minute burst data, which overlap well with the overall session burst data, in Fig S1, would seem to indicate that burst size is increased prior to satiation signals coming online (i.e., in the early part of the session). Perhaps looking at dynamic changes in lick rate (or burst size) in the 3 min bins across the entire session would clarify. The burst size does not appear to be reduced until the second phase of the experiment. This may be a limitation of interpreting burst size as a function of reward and burst number as a function of motivation. Of course, it is impossible to know whether this is simply due to an accumulation of drug or something about the pre and post injection procedure. I suppose, at the very least, if the drug at these doses was increasing the rewarding value of sucrose, it did not drive overconsumption (intake was pretty stable across the days).

Line 45-46 is a little unclear. Burst number cannot be separated from orosensory cues in this design.

7. PLOS authors have the option to publish the peer review history of their article (what does this mean?). If published, this will include your full peer review and any attached files.

Reviewer #1: **Yes: **Dr. Fabien Naneix

Reviewer #2: No

---

## [Author Response · Author response to Decision Letter 1]

26 Dec 2020

Please notice that in the Revised Manuscript with Track Changes the new text is highlighted with a light grey background. Numbers in square brackets refer to citations included in the References of the submitted paper.

Reviewer #2: The paper is greatly improved by reframing of it in terms of examining the “immediate” effects of the imipramine on sucrose licking patterns in the absence of interposed post-session administrations, as opposed to invoking some examination of learning or memory. In general, the discussion is a lot more cautious about mechanism, which is appropriate given the design and data. I still have a couple of outstanding concerns.

Response: We thank the Reviewer for this comment.

Perhaps the authors could add some indication of which sessions are included in part 1 and 2 in the table. But, in general, the table helps clarify the protocol.

Response: Table 1 was integrated with the requested indications (clean ms: page 4; rev. man. with track changes: page 5).

I would like to see more of the within session data (plotted across time). Per my previous review, I thought it would be important to examine burst size in addition to be able to interpret the burst number data across the session. The authors now present the first 3 min burst size in the supporting information, but not beyond that. The authors appear to favor the interpretation that imipramine reduces behavioral motivation or increases satiation. But for the 10 mg dose (and maybe the 5), the data could just as easily be interpreted as increase in “hedonic value” based on the framework provided by the authors in the introduction. In fact, the first 3 minute burst data, which overlap well with the overall session burst data, in Fig S1, would seem to indicate that burst size is increased prior to satiation signals coming online (i.e., in the early part of the session). Perhaps looking at dynamic changes in lick rate (or burst size) in the 3 min bins across the entire session would clarify. The burst size does not appear to be reduced until the second phase of the experiment. This may be a limitation of interpreting burst size as a function of reward and burst number as a function of motivation. Of course, it is impossible to know whether this is simply due to an accumulation of drug or something about the pre and post injection procedure. I suppose, at the very least, if the drug at these doses was increasing the rewarding value of sucrose, it did not drive overconsumption (intake was pretty stable across the days).

Response: 

1) The within-session time-course of burst size of the first three sessions (i.e. of the sessions for which it is reported the within-session time-course of burst number, as requested by the Reviewer in the previous run of this reviewing process) was provided as Supporting information (S2 Fig, see also S1 Data for the individual numbers). These data do not seem to show changes in burst size in the course of the session. However, an important limitation to the possibility to make a meaningful interpretation of these data comes from the fact that, as the sessions proceeds, the subjects stop licking, thus resulting in empty cells late in the session. Firstly, this leads to an exclusion of too many subjects from the repeated measures ANOVA (which does not allow empty cells). Maybe more importantly, the lack of a relevant number of subjects in the late session time bins – which are not necessarily the same in different time bins – brings as a consequence that the comparisons between the different time bin values early and late in the session are heavily affected by differences between subjects (which are inextricably confounded with possible changes in time within the session). The data relative to the within-session time course of burst size are described in the Results section (clean ms: lines 237-244; rev. man. with track changes: lines 244-251) and mentioned in the Discussion (clean ms: 331-332; rev. man. with track changes: lines 339-340). We wish to stress that this experiment was designed in order to consider the whole session burst size as the relevant measure, as highlighted in the Discussion (clean ms: 385-397; rev. man. with track changes: 394-406). Indeed, the best way to investigate changes of burst size in the course of ingestion involves the recording of the data in relation to the progression of a meal, rather than in relation of a fixed time session (e.g. [10]).

2) We absolutely agree with the Reviewer's considerations in regard of the interpretative issues. As acknowledged by the Reviewer in the preliminary remark, in general, the previously submitted text (i.e. the first revision) is a lot more cautious in the formulation of the proposed interpretations. In particular, following this Reviewer's advice, in the first revision we acknowledged the limitations in regard of the interpretation of the differences between the first and the second part of the experiment. Moreover, all the statements about the interpretation of the results in terms of reduced motivation were formulated in cautious terms (e.g. “suggest”: clean ms: 325, 400; rev. man. with track changes: line 333, 409; “possibly due”: clean ms: 420; rev. man. with track changes: 429). However, although the possibility that the increased burst size might be due to an increased hedonic impact was not dismissed in the previous version (please see final part of the conclusive remarks of the discussion, unmodified), we must admit that the formulation of the text “downplayed” this interpretation. Thus, in the currently submitted text we made these changes/integrations:

-“is not” was changed into “does not appear to be” (clean ms: 378-379; rev. man. with track changes: 386-387).

- It was added the following sentence: “Nonetheless, further experiments are necessary to determine whether the effect of imipramine on burst size might be accounted for by an increased hedonic impact.” (clean ms: 397-398; rev. man. with track changes: 406-407).

Line 45-46 is a little unclear. Burst number cannot be separated from orosensory cues in this design.

Response: We agree with the reviewer. Indeed, one of the central interests in our previous work on licking is about the interpretation of the within-session changes of burst number in response to changes in sucrose concentration or to treatments which might affect the hedonic response to taste (e.g. [4,22,26,39]). To clarify the point indicated by the Reviewer in the Introduction, with added a note in parentheses (clean ms: lines 49-50; rev. man. with track changes: lines 52-53). This required a rearrangement of the sentence order within the paragraph (rev. man. with track changes: lines 43-54). In particular, we left unchanged the statement “The number of licking bursts […] is particularly sensitive to stimuli […] which do not involve the orosensory contact with the reward.”, and added a cautionary note: “(But notice that this measure does respond to orosensory cues, albeit in a less straightforward fashion than burst size.)”.

***

Finally, we wish to thank both the Editor and the Reviewers for their helpful and constructive comments.

---

## [Editor Report · Decision Letter 2]

4 Jan 2021

Further characterization of the effect of the prototypical antidepressant imipramine on the microstructure of licking for sucrose

PONE-D-20-29592R2

Dear Dr. D'Aquila,

We’re pleased to inform you that your manuscript has been judged scientifically suitable for publication and will be formally accepted for publication once it meets all outstanding technical requirements.

Kind regards,

James Edgar McCutcheon, Ph.D.

Academic Editor

PLOS ONE
---

## [Editor Report · Acceptance letter]

7 Jan 2021

PONE-D-20-29592R2 

Further characterization of the effect of the prototypical antidepressant imipramine on the microstructure of licking for sucrose 

Dear Dr. D'Aquila:

I'm pleased to inform you that your manuscript has been deemed suitable for publication in PLOS ONE. Congratulations! Your manuscript is now with our production department. 

Kind regards, 

on behalf of

Dr. James Edgar McCutcheon 

Academic Editor

PLOS ONE